# Photonic chip-based soliton frequency combs covering the biological imaging window

Maxim Karpov[1], Martin H.P. Pfeiffer[1], Junqiu Liu[1], Anton Lukashchuk[1] & Tobias J. Kippenberg[1]

Dissipative Kerr solitons (DKS) in optical microresonators provide a highly miniaturised, chip-integrated frequency comb source with unprecedentedly high repetition rates and spectral bandwidth. To date, such frequency comb sources have been successfully applied in the optical telecommunication band for dual-comb spectroscopy, coherent telecommunications, counting of optical frequencies and distance measurements. Yet, the range of applications could be significantly extended by operating in the near-infrared spectral domain, which is a prerequisite for biomedical and Raman imaging applications, and hosts commonly used optical atomic transitions. Here we show the operation of photonic-chip-based soliton Kerr combs driven with 1 micron laser light. By engineering the dispersion properties of a $Si_3N_4$ microring resonator, octave-spanning soliton Kerr combs extending to 776 nm are attained, thereby covering the optical biological imaging window. Moreover, we show that soliton states can be generated in normal group–velocity dispersion regions when exploiting mode hybridisation with other mode families.

[1] École Polytechnique Fédérale de Lausanne (EPFL), Laboratory of Photonics and Quantum Measurements (LPQM), 1015 Lausanne, Switzerland. Correspondence and requests for materials should be addressed to T.J.K. (email: tobias.kippenberg@epfl.ch)

Microresonator-based frequency combs are optical frequency combs generated from a continuous-wave (CW) laser via parametric four-wave mixing processes in high-$Q$ microresonators[1]. They have attracted significant interest owing to their compactness, CMOS-compatible fabrication, and ability to operate with repetition rates in the gigahertz to terahertz range with a broad spectral bandwidth[2]. It was recently demonstrated that such CW laser-driven microresonators can support the spontaneous formation of dissipative Kerr solitons[3] (DKS)—self-organised stable intracavity pulses relying on the double balance between dispersion and nonlinearity[4], and parametric gain and cavity losses—which provide a route to fully coherent optical frequency combs whose spectral bandwidth can be significantly broadened via soliton-induced Cherenkov radiation[5–7]. Such microresonator-based DKS combs were demonstrated in several platforms[3,5,8,9,12] and have already been successfully applied in optical coherent communications[10], dual-comb spectroscopy[11], implementation of the microwave-to-optical link via self-referencing[13,14] and most recently in dual-comb distance measurements[15,16]. Yet, to date, well-identified and single DKS have only been attained in the telecommunication bands (around 1550 and 1300 nm), where a wide variety of DKS-supporting microresonator platforms have been developed.

Importantly, a large class of new applications of biomedical nature can be accessed with DKS-based comb sources if they can operate in the short-wavelength region of the near-infrared (NIR) domain ranging from 0.7 to 1.4 μm wavelength. This spectral region is used for biological and medical imaging due to the highest penetration depth in biological tissues [17]. Optical spectroscopy, Raman spectro-imaging and optical coherence tomography (OCT) techniques operating in this wavelength range serve as a non-invasive tool for the structural and chemical analysis of various biological samples, including retinal and choroidal structures or tumour formations[18–20]. These biomedical imaging techniques could benefit from employing optical frequency combs as light sources due to their coherence and high power per comb line[21], as well as from using dual-comb approaches allowing to interrogate the broad spectral bands with a single photodetector[22]. A specific example of dual-comb-based spectroscopy in the NIR is coherent anti-Stokes Raman spectroscopy (CARS)[23], which potentially can utilise high repetition rates of the DKS combs for vastly increased acquisition rate, enabling real-time CARS imaging. Furthermore, the luminescence-free anti-Stokes response can benefit from the larger Raman cross section and reduced focal spot size at shorter wavelengths, facilitating phase matching. Such dual-DKS-comb CARS might be able to provide ultrafast multispectral in vivo imaging for chemical, biological and medical purposes. Equally important, a range of other applications requiring a stabilised operation can be accessed and improved by NIR DKS-based combs. The NIR domain hosts optical frequency standards in alkali vapours (e.g.,[87]Rb, [133]Cs), necessary to realise chip-scale optical atomic clocks with enhanced precision[24], or employed in DKS-comb-based calibration for astronomical spectrometers[25].

Despite the large number of promising applications of NIR DKS-based combs, such sources with full integration have so far not been developed. Although attempts to generate Kerr combs in the NIR and visible domain have been made[26–30], they resulted in relatively narrow and incoherent combs, hence soliton formation has not been achieved. The generation of NIR or visible soliton combs is compounded by the increased normal group–velocity dispersion (GVD) of the materials due to the electronic bandgap in UV, and the increased scattering losses and sensitivity to the resonator waveguide dimensions which require higher precision in dispersion engineering and fabrication processes. Moreover, as shown recently, competition between Raman and Kerr effects in the NIR or visible domains can inhibit soliton formation[31].

In this work, we demonstrate that $Si_3N_4$ microresonators can overcome these challenges and the DKS-based frequency combs can be generated with a 1060 nm CW laser, thereby allowing access to the optical wavelength window for biological imaging (0.7–1.4 μm). The typical signatures[3,32,33] of DKS in microresonators, including low-phase-noise operation, Raman-induced red spectral shift, soliton switching dynamics and the bistability-related double-peak response of the microresonator system are observed and provide unambiguous identification of the DKS states. Equally important, we demonstrate the formation of octave-spanning soliton states in this wavelength region by exploiting the coherent dispersive wave emission for efficient spectral broadening towards visible wavelengths. Finally, we report the soliton formation in hybridised microresonator modes, which represents an alternative approach to extend the DKS operation further into the visible domain where normal GVD is usually dominant. The DKS states are achieved in the region of an avoided modal crossing, where the strong interaction between resonator modes leads to locally enhanced anomalous GVD allowing DKS formation. We show that the bandwidth of such soliton states is highly sensitive to the pumped resonance within the interaction region—a behaviour contrasting the behaviour in the absence of modal crossings.

## Results

**Device design and characterisation.** We employed the silicon nitride ($Si_3N_4$) microresonator platform, which is a well-developed and extensively used basis for on-chip nonlinear and quantum photonics due to a number of advantages such as CMOS-compatibility, high effective nonlinearity, negligible two-photon absorption and wide transparency window spanning from visible to mid-infrared region[34–36]. Recent advances in the fabrication processes have enabled the fabrication of crack-free and low-loss $Si_3N_4$ waveguides with void-free coupling gaps, which guarantee high-$Q$ resonators with well-controllable properties[7,37–40]. An important advantage of this is the ability to engineer the dispersion properties of the microresonators by compensating the material dispersion with the geometry-dependent waveguide dispersion contribution[5–7]. In the context of microresonator-based Kerr frequency combs, the dispersion properties are often expressed through the frequency deviations from an equidistant grid for a certain pumped mode $\omega_0$ in the relative mode index ($\mu$) representation: $D_{int}(\mu) = \omega_\mu - (\omega_0 + D_1\mu) = \sum_{i>1} D_i\mu^i/i!, i \in \mathbb{N}$, where $\omega_\mu$ is the angular frequency of the cavity resonance, and $\frac{D_1}{2\pi}$ is the free-spectral range (FSR) of a microresonator[3]. For bright DKS generation, it is generally required to achieve anomalous GVD: $D_2 > 0$, which can be especially challenging at short wavelengths.

In this work, we used $Si_3N_4$ microrings with FSR of ~1 THz (radius ~23 μm, see Fig. 1b), which were fabricated using the photonic Damascene process[38]. The resonator waveguide width was varied from 1.3 to 1.5 μm, and the height was targeted at 0.74 μm (with process-related variations in the order of 20 nm). The geometry was chosen based on FEM simulations (see Fig. 1c), in order to ensure anomalous GVD of the fundamental TM mode around the pumping wavelength of 1060 nm (we note that the fundamental TE mode has normal GVD at the same wavelength in all fabricated geometries). The bus waveguide has a pulley-style coupling section with altering width ranging from 0.3 μm at the in-coupling part (see Fig. 1b - C) to 0.65 μm at the out-coupling part (see Fig. 1b - A, B). The coupling section was designed to guarantee broadband and good ideality coupling, reaching ~0.85 at 1 μm according to our simulations[41], which can be further

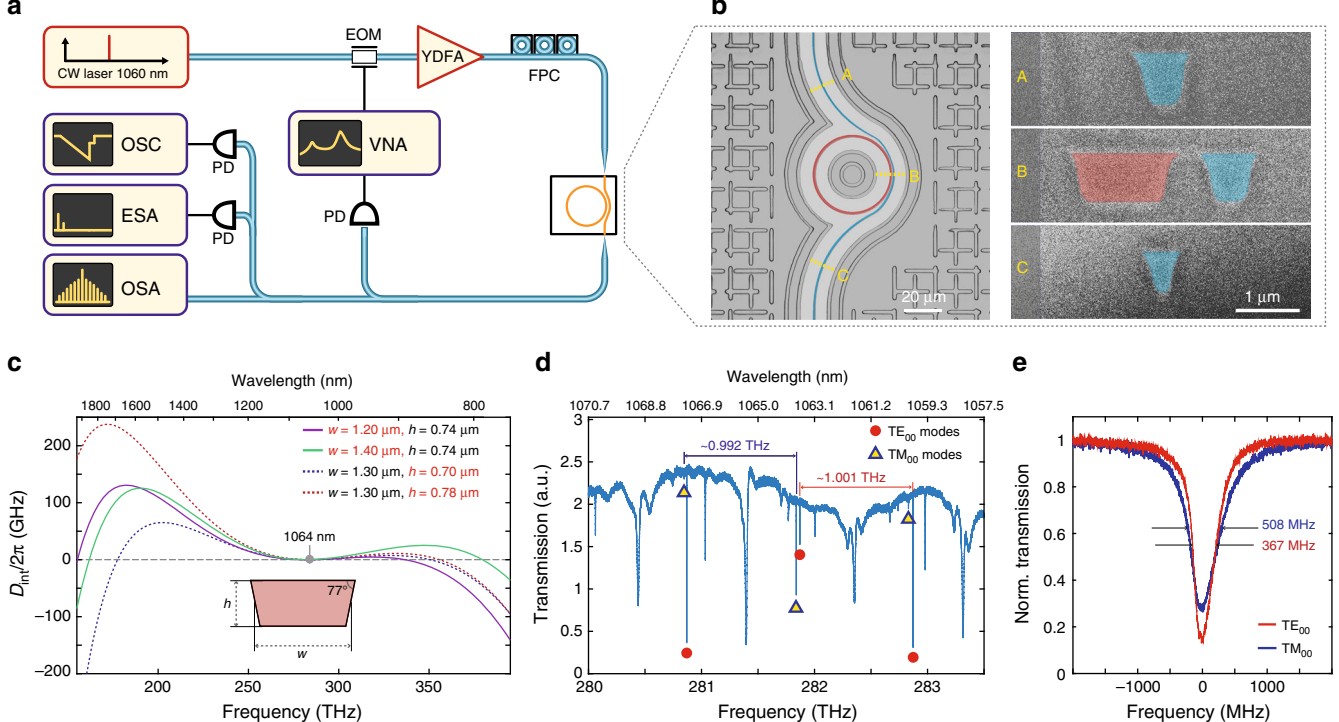

**Fig. 1** $Si_3N_4$ microresonators for soliton frequency comb generation in the NIR. **a** Set-up schematic used for dissipative Kerr soliton generation and characterisation: a tunable external-cavity diode laser with a centre wavelength of 1050 nm is used as a seed, YDFA – ytterbium-doped fibre amplifier, FPC – fibre polarisation controller, VNA – vector network analyser, EOM – electro-optical phase modulator, PD – photodiode, OSC – oscilloscope, ESA – electrical spectrum analyser, OSA – optical spectrum analyser. **b** Left: optical microscope image of the 1-THz microring resonator (highlighted in red) with a pulley-style bus waveguide (blue). Right: scanning electron microscope images of the resonator and bus waveguide cross sections obtained via focused ion beam cuts at different positions (A–C) marked on the left image. **c** Simulated integrated dispersion profiles ($D_{int}/2\pi$) for $TM_{00}$ mode of resonator waveguides having different heights of 0.70, 0.74 and 0.78 μm, widths of 1.2, 1.3 and 1.4 μm and fixed sidewall angle of 77° (see more details on dispersion engineering in Supplementary Note 1). **d** Transmission trace of the 1-THz microresonator shown in (**b**). The two fundamental mode families ($TE_{00}$ and $TM_{00}$) can be distinguished based on their free spectral ranges (0.992 THz for $TM_{00}$ and 1.001 THz for $TE_{00}$), and are marked with red and blue shapes correspondingly. Other resonances correspond to higher order modes with comparably lower Q-factors. **e** Linewidth measurements of the fundamental mode families. The frequency axis was calibrated using a fibre-loop cavity. Typical loaded linewidth of the modes is ~400 MHz for $TE_{00}$ and ~500 MHz for $TM_{00}$

optimised to reduce parasitic losses. The waveguides have inverse tapered mode converters (reducing the waveguide width down to ~150 nm) at the input and output ends, providing <3 dB coupling loss per facet at 1 μm.

A typical transmission trace of the fabricated devices is shown in Fig. 1d. The comparably large width of the microring waveguide induces significant overmoding of the resonator. Two fundamental modes—$TE_{00}$, $TM_{00}$ can be easily identified due to different FSRs (0.992 THz for $TM_{00}$ and 1.001 THz for $TE_{00}$) provided by the non unity aspect ratio. Different coupling of the TE and TM fundamental modes over the measured range in Fig. 1d is attributed to the polarisation plane rotation in the optical fibre before coupling to the on-chip bus waveguide. Once the polarisation is properly adjusted at a certain wavelength for the resonance of a particular mode family, one can measure its Q-factor. Both the fundamental modes have comparable loaded linewidths of 370–550 MHz, corresponding to the Q-factors of $0.55–0.75 \times 10^6$ (see Fig. 1e).

**Dissipative Kerr solitons at 1 μm.** The experimental set-up for the DKS generation in $Si_3N_4$ microresonators is shown in Fig. 1a. The light from a tunable 1060 nm CW diode laser (Toptica CTL 1050) is amplified with an ytterbium-doped fibre amplifier and coupled to the chip through a lensed fibre. The output signal was collected with another lensed fibre, and its spectral and noise

characteristics were analysed. We also employed a recently developed soliton probing technique[33], which uses a phase-modulated pump and a vector network analyser to retrieve the system response, allowing us to unambiguously identify DKS formation and track the detuning of the pump from a cavity resonance.

It is well-known, that the CW-driven nonlinear cavity can support DKS states when operating in the effectively red-detuned regime ($\omega_p - \omega_0 = 2\pi\delta > 0$, where $\omega_p$ and $\omega_0$ are the angular frequencies of the pump laser and pumped resonance, respectively)[3]. A standard approach for accessing such states is the laser-tuning method[3,33], where the pump laser is swept over the resonance from the blue to the red side, and is stopped at a certain pump-resonance detuning ($2\pi\delta > \frac{\sqrt{3}}{2}\kappa$, where $\kappa/2\pi$ is the total microresonator linewidth) supporting soliton formation. For the samples used in our work, it is sufficient to apply this standard approach, rather than complex techniques such as power kicking[5] or fast frequency modulation[42]. This can be explained with the relatively small pump powers used for soliton generation. They allowed to avoid the region of transient chaos[43], causing the elimination of intracavity pulses during the transition to a stable soliton state. In this case, thermal effects associated with the change of intracavity power do not significantly affect the pump-cavity detuning, and the system stays within the soliton existence range after the DKS are formed[44].

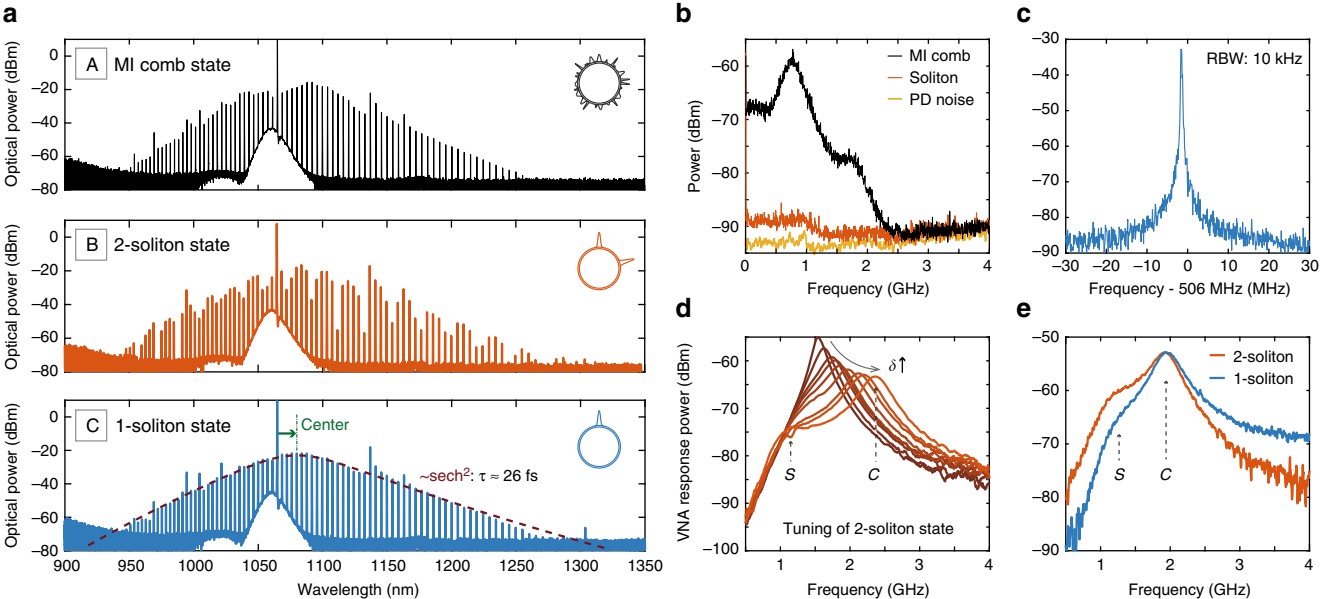

**Fig. 2** Dissipative Kerr solitons at 1 µm and their characterisation. **a** Optical spectra of the modulation-instability (MI) comb state (A) and two soliton states (B and C) obtained in a 1-THz Si$_3$N$_4$ microresonator (pump is located at around 1065 nm). The transition from two-soliton state (B) to single-soliton state (C) was obtained by backward tuning technique of the pump laser. Insets show the estimated positions of the DKS in corresponding states. The single-soliton state was fitted with the sech$^2$ envelope (dashed dark blue) for an estimated duration of 26 fs. The green arrow shows the Raman-induced red spectral shift of the soliton spectrum with respect to the pump line. **b** Intensity noise of MI comb state (black), soliton states (red) shown in Fig. 1 **a** and the noise floor of the photodiode (PD) used for measurements (yellow). **c** Heterodyne beatnote of the soliton comb line around 1050 nm with a second CW laser. **d** System response evolution in the two-soliton state shown in **a**, when increasing the pump-cavity detuning ($\delta$). The positions of characteristic $\mathcal{C}$-resonance and $\mathcal{S}$-resonance are indicated with C and S letters correspondingly. **e** System response evolution when the DKS state is switched from two-soliton to single-soliton state. The amplitude of the $\mathcal{S}$-resonance has decreased, because the number of intracavity solitons is reduced

We apply this technique with an on-chip pump power of ~1 W to the resonances of fundamental TM mode family, which should provide anomalous GVD at around 1060 nm. Low tuning speeds on the order of few GHz/second (i.e., using the laser cavity piezo) were chosen and enabled simultaneous monitoring of the cavity state by measuring the system response signal, as well as an optical spectrum of the output light. The cavity reveals modulation instability and noisy Kerr comb formation while the pump laser is on the blue side of the resonance (see Fig. 2 (a, top)). Upon further pump tuning, the transition to the soliton regime is accompanied by a change of the optical spectrum to a secant hyperbolic-like shape, and the appearance of a double-peak structure in the system response representing the coexistence of the soliton ($\mathcal{S}$-resonance) and the CW-background ($\mathcal{C}$-resonance) components inside the cavity. To explore the soliton existence range of the generated DKS state, response measurements were also carried out, while tuning the pump laser towards longer wavelength corresponding to the increase of effective detuning (Fig. 2d). The expected shift in the position of $\mathcal{C}$-resonance (which has been shown to indicate the effective detuning of the system) to higher frequencies is clearly seen, and essentially reproduces the dynamics of similar response signals measured for DKS at 1550 nm[33]. Finally, the transition to the soliton regime has also been verified by the drastic reduction of output-light-intensity-noise (see Fig. 2b), and a narrow heterodyne beatnote of the selected comb line at around 1050 nm with another CW diode laser (Fig. 2c).

We also demonstrated that the obtained soliton states can experience switching by applying the recently reported backward tuning technique, which relies on the thermal nonlinearity of microresonators, and allows the number of DKS circulating inside the cavity to be changed in a robust and controllable way[33]. Figure 2a shows the switching from a two-soliton state to a single-

soliton state. The switching has also been confirmed with the response measurements as shown in Fig. 2e. A decrease in the amplitude of the soliton-number-related $\mathcal{S}$-resonance of the response indicates the reduction in intracavity number of pulses, while the cavity-related $\mathcal{C}$-resonance is almost unchanged. By fitting the spectrum of the final single-soliton state with a sech$^2$ envelope, the soliton duration can be estimated from its 3-dB bandwidth as 26 fs. We also note the significant soliton red spectral shift (~4.1 THz in the present case) with respect to the pump line, which is mainly attributed to the Raman effect and observed for all DKS states [32,45].

**Octave-spanning soliton states.** Reaching the octave-spanning operation of DKS is an important step in the development of Kerr frequency combs, as it enables the common $f - 2f$ scheme for the offset frequency detection and self-referencing required by multiple applications in optical frequency metrology and low-noise microwave synthesis[46]. Octave-spanning DKS states have only very recently been demonstrated experimentally[6,7]. Here, we demonstrate that DKS-based combs operating at 1 µm can also be engineered to have octave-spanning bandwidths, despite operating close to the normal GVD region in the Si$_3$N$_4$ platform.

We used a microresonator with the same FSR of 1 THz as in the previous section, having a waveguide geometry of $1.30 \times 0.74$ µm, which was designed to maintain the low anomalous GVD and satisfy phase-matching conditions at around 800 nm for dispersive wave formation[5]. This spectral region is particularly interesting due to the presence of optical frequency standards based on the two-photon Rb transitions[47], which can be used for comb referencing. We applied the aforementioned low-speed tuning technique with an estimated on-chip power of ~800 mW

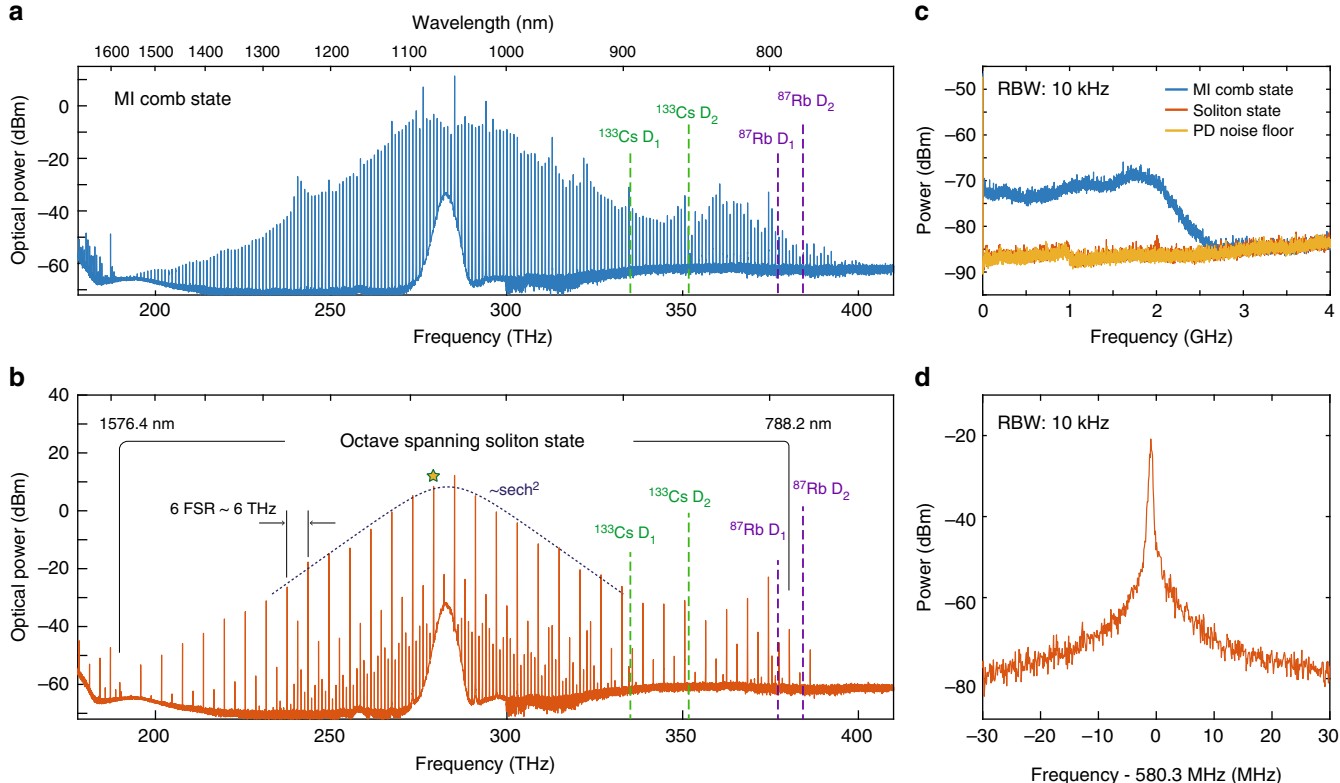

**Fig. 3** Octave-spanning microresonator-based dissipative Kerr soliton state in the biological imaging window. **a** Spectrum of a noisy Kerr comb state obtained with ~800 mW on-chip power (pump is located at around 1051.5 nm), from the microresonator with optimised dispersion for octave-spanning operation. Green and purple dashed vertical lines indicate the spectral locations of Cs and Rb optical atomic transitions. **b** Spectrum of the octave-spanning soliton state obtained in the same microresonator. The spectrum is fitted with $sech^2$ envelope (dashed dark blue), from which a soliton duration of 17 fs is inferred. The set of enhanced lines below 190 THz corresponds to the second diffraction order of optical spectrum analyser diffraction grating, and are thus artefacts. **c** Intensity noise of the octave-spanning MI comb state (blue) and soliton state (red) shown in (**a**). Yellow trace shows the noise floor of the photodetector used for measurements. **d** Heterodyne beatnote of the soliton comb line (marked with a star in panel (**b**)) with a second CW laser

to achieve the formation of the noisy comb (see Fig. 3a) followed by the soliton state (Fig. 3b).

The resulting spectrum of the soliton state spans over an octave ranging from 776 to 1630 nm (>200 THz). As expected, it is significantly extended towards shorter wavelengths due to the emission of the dispersive wave via soliton-induced Cherenkov radiation at 800 nm[48]. The 3-dB bandwidth of the spectrum fitted with the $sech^2$ envelope is estimated as 18 THz, which corresponds to ~18-fs pulse. A peculiar shape of the DKS state, consisting of several soliton-like spectra with different FSR is attributed to the formation of a multiple-soliton state represented by an ordered co-propagating DKS ensemble–soliton crystal[44,49]. Such soliton crystals are typically formed in the presence of strong local spectrum deviations caused by intermode interactions among transverse mode families (avoided mode crossings), and in contrast to single-soliton states, they are featuring high conversion efficiency (owing to the high number of intracavity pulses), which in the present case approached 50%.

Similar to the DKS states demonstrated in the previous section, the presented state is also characterised by a low-noise perfomance with a strongly suppressed intensity noise in comparison to the noisy modulation-instability (MI) comb (see Fig. 3c), and a narrow heterodyne beatnote of the generated comb lines with another CW laser (Fig. 3d).

**Dissipative Kerr soliton states in hybridised modes.** The behaviour of Kerr combs, and in particular DKS states are to a large extent defined by the dispersion properties of the cavity.

One of the key requirements for bright DKS formation is the anomalous GVD of the microresonator, which can be achieved by overcompensating the normal material dispersion with waveguide dispersion contribution. At short wavelengths, however, the increased normal material GVD can represent a significant issue (e.g. for $Si_3N_4$ in the visible domain), as it can hardly be compensated for the fundamental guided modes, thus hindering the bright DKS formation. Apart from the global dispersion landscape comprised of the material-related and waveguide-related components, the dispersion properties also include spectrally localised dispersion modifications (avoided mode crossings, AMX) that are typically caused by the formation of guided hybridised modes, which can appear due to the interaction of different transverse mode families[50]. Although being spectrally localised, such AMX-s can lead to complex and diverse effects on the dynamics of the DKS states, such as dispersive wave formation, soliton recoil, temporal soliton ordering, appearance of quiet operation points and intermode soliton breathing[3,44,49,51,52]. Moreover, AMX has been reported to allow the formation of mode-locked states consisting of dark pulses in resonators with normal GVD[53].

In this section, we demonstrate that localised strong anomalous GVD of the hybridised modes around AMX can be directly employed for bright soliton generation irrespective of the global dispersion profile. The effect is schematically explained in Fig. 4a, where the simulated integrated dispersion is plotted (solid lines) for several modes of the 1-THz resonator ($0.74 \times 1.45$ µm, sidewall angle 77°). We consider the TE fundamental mode family, which according to our simulations has normal GVD

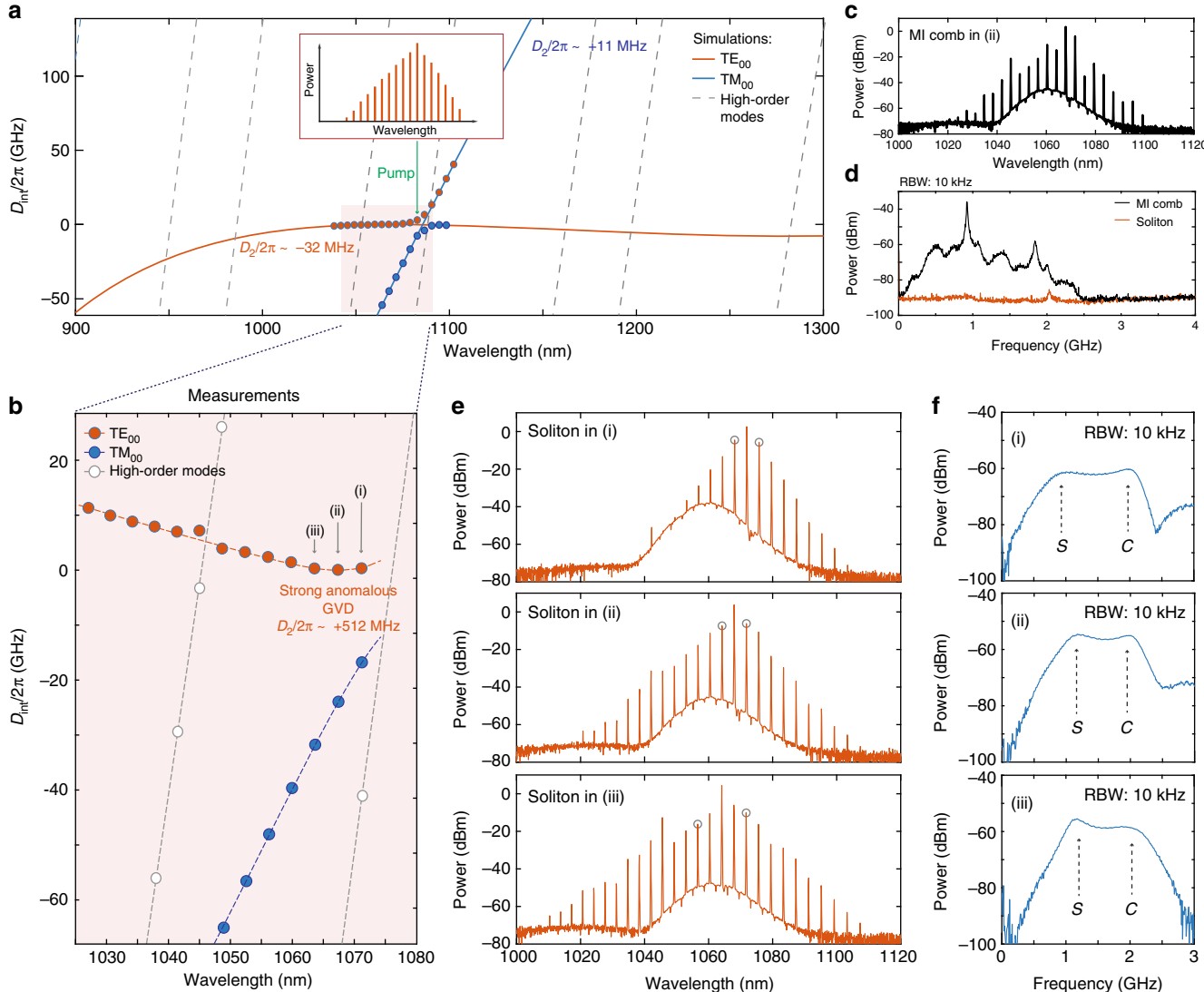

**Fig. 4** Dissipative Kerr solitons in hybridised modes. **a** Scheme of the avoided mode crossing formation. Solid and dashed lines show the simulated integrated dispersion for the fundamental and high-order modes of 1-THz $Si_3N_4$ microresonator with dimensions of $1.45 \times 0.74\,\mu m$. Red and blue circles schematically indicate the dispersion profile of the hybridised modes forming AMX. **b** Measurements[54] of the mode structure of a 1-THz $Si_3N_4$ microresonator with the dimensions of $1.45 \times 0.74\,\mu m$ simulated in (**a**). Mode families are distinguished based on their FSRs. General dispersion trends cannot be faithfully identified due to the bandwidth limitations of our measurement setup, however a strong local anomalous group–velocity dispersion for three consecutive resonances (i), (ii), (iii) of the $TE_{00}$ mode family above 1064 nm can be observed. Dashed lines fit the integrated dispersion of different modes within the measurements range. **c** Optical spectrum of a noisy comb state, obtained in resonance (ii) from (**b**). **d** Intensity noise measurements of the noisy modulation-instability comb (MI comb, black) and the dissipative Kerr soliton (soliton, red) states, obtained by pumping the resonance (i) from (**b**). **e** Optical spectra of the DKS states obtained by pumping resonances (i), (ii), (iii) from **b**. Grey circles indicate the positions of primary comb lines. **f** Response measurements of DKS states represented in (**e**). The positions of characteristic $\mathcal{C}$-resonance and $\mathcal{S}$-resonance are indicated with $C$ and $S$ letters, correspondingly

$\left(\frac{D_2}{2\pi} = -32\,\text{MHz}\right)$, and note that it has multiple modal crossings with other modes. The frequency degeneracy of different mode families that appears at such modal crossings can lead to mode interaction mediated by scattering processes in the microresonator (see Supplementary Note 3) and induce the formation of AMX. The dispersion profile of one such AMX is schematically shown in Fig. 4a (circles) to highlight its strong deviation from the simulation results obtained in the absence of scattering. Figure 4b shows the measured[54] mode structure of one of the fabricated samples with the same geometry ($0.74 \times 1.45\,\mu m$), where the formation of the described AMX is experimentally observed to be close to the simulated spectral position of 1080 nm

(see Fig. 4b). Mode-interaction-induced strong resonance shifts cause a dramatic change in the local GVD of the TE mode family turning it from normal with $\frac{D_2}{2\pi} = -32\,\text{MHz}$ (simulated) to highly anomalous, with $\frac{D_2}{2\pi}$ reaching 510 MHz (measured). We therefore evidence an AMX-induced change in the local GVD.

Driving the modes with such a strong anomalous GVD (when $\sqrt{\frac{\kappa}{D_2}} < 1$) should result in the formation of a natively mode-spaced comb[55,56], whose primary lines appear 1 FSR away from the pump due to the closely located MI gain peaks. Previous works have reported that such combs can appear directly in a mode-locked regime[57] which, however, is in contrast with our observations. In the experiments, we again used the same pump tuning technique as

in previous sections, applied to hybridised modes forming AMX. Using the system response and intensity noise measurements, we observe the standard soft-excitation-route of a Kerr comb formation, which includes the appearance of primary comb lines, development of chaotic modulation instability (MI, Fig. 4c) and the transition to a soliton regime (Fig. 4e), which was verified by the appearance of a characteristic dual-peak system response and low-intensity noise (Fig. 4e,f, see also Supplementary Note 2 for more details). Similar behaviour was observed over three consecutive resonances (cases I, II, III in Fig. 4e), where the corresponding soliton states were generated. An interesting observation can be made regarding the spectral bandwidth of the obtained soliton states. As a result of the localised character of the anomalous GVD of the hybridised modes, the actual value of the dispersion terms (and particularly $\frac{D_2}{2\pi}$ term) varies from one resonance to another, leading to the different effective detunings for generated comb lines and altering the resulting spectral width. We also note here, that due to the contribution of higher-order dispersion terms around AMX, the obtained DKS spectra cannot be faithfully fitted with the sech$^2$ envelope.

The demonstrated soliton states generated in the hybridised modes by exploiting their strong anomalous GVD can represent an alternative way to deterministically generate soliton-based optical combs in arbitrary wavelength regions. This approach can be especially useful for the Si$_3$N$_4$ platform presented here to generate soliton states operating at 780 nm or even further into the visible domain, where normal GVD cannot be efficiently compensated with resonator waveguide geometry, but can be locally altered using e.g., predesigned or thermally controlled AMXs[58].

## Discussion

In conclusion, we show the photonic-chip-integrated soliton-based optical frequency comb sources driven with 1 µm pump source. The spectra of the demonstrated DKS states are able to span over an octave and cover the common optical frequency standards in alkali vapours, as well as a significant part of the NIR biological imaging window. Moreover, we show that DKS states can be generated in hybridised microresonator modes around avoided mode crossings by directly exploiting their localised anomalous GVD, which represents an alternative approach for the generation of DKS combs in regions with strong normal GVD (e.g. at shorter wavelength in Si$_3$N$_4$ and in other materials). From a broader perspective, our work gives strong evidence of the technological readiness of the Si$_3$N$_4$ platform for soliton-based operation in the NIR domain around 1 µm, including comparably good quality factors and the means of dispersion engineering, which makes it a highly promising candidate for multiple biological and other applications in this spectral window, including OCT and dual-comb CARS. Finally, we would like to draw the reader's attention to the other work on the generation of DKS states centred at 1064 and 780 nm in fibre-coupled silica microdiscs[59].

**Data availability**. The code and data used to produce the plots within this paper are available at 10.5281/zenodo.1149180. All other data used in this study are available from the corresponding authors upon reasonable request.

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

## Acknowledgements

This publication was supported by Contract W31P4Q-14-C-0050 from the Defence Advanced Research Projects Agency (DARPA), Defence Sciences Office (DSO); by the Air Force Office of Scientific Research, Air Force Material Command, USAF under Award No. FA9550-15-1-0099; We also acknowledge the support from the European Space Technology Centre with ESA Contract No. 4000116145/16/NL/MH/GM and the support from the European Union's FP7 programme under Marie Sklodowska-Curie Initial Training Network grant agreement No. 607493. $Si_3N_4$ samples were fabricated and grown in the Center of MicroNanoTechnology (CMi) at EPFL.

## Author contributions

M.K. designed and performed experiments and analysed the data. M.P. designed the samples and performed dispersion simulations. A.L. assisted with dispersion simulations. J.L. implemented ideality simulations. M.K. wrote the manuscript with an input from M. P., J.L., A.L., T.J.K. T.J.K. supervised the project.

## Additional information

**Competing interests:** The authors declare no competing interests.

