## [Peer Review File(PDF 770 kb) · Nature Communications]

Reviewers' comments:

Reviewer #1 (Remarks to the Author):

This paper written by Maxim et al reports on the dissipative soliton on silicon nitride microring for NIR and visible wavelength. This is a good integrated platform to generate comb with biological range. The octave spectrum is suitable for the stabilization of the comb by $f-2f$ and some short wavelength comb lines benefit the optical clock based on the Rb transition. But I think the properties of the comb in this paper do not reach the Nature communications level:

1. In Figure 1d, the transmission is not good, especially for TM mode. It means the coupling condition is not suitable for TM mode and pump power coupled into the cavity is low. So it is not best choice that the author select TM mode as pump.

2. In Figure 2, authors got the three and double pulses, not single pulse. This kind of comb has less lines and is not best source for bio-spectroscopy cause the FSR is already large.

3. In Figure 3a, although authors obtained octave spectrum, but both side comb lines are very weak and less. The $f-2f$ is very difficult and low efficient. According to the spectrum shape it is like breather soliton, not single pulse.

4. The concept of hybridized mode is new, but it is difficult to control dispersion compared with mode interaction [Nat. Photon. 9, 594–600 (2015)].

So unfortunately this manuscript is not recommended at its present form and authors can design new chip again and optimize the properties of the comb.

Reviewer #2 (Remarks to the Author):

This paper describes about the experimental demonstration of octave spanning soliton generation centered at 1050 nm wavelength. It is based on their previous study (Ref. 34), but extending their wavelength to 1050 nm regime by conducting careful dispersion engineering. Due to the tradeoff between the small FSR and large bandwidth, the use of a small cavity with large FSR of 1-THz allows the authors to obtain broad spectrum width, compare to the similar work conducted by Vahala et al. (ref. 58) that uses a cavity with 40-GHz FSR. As a result of the large bandwidth, this work achieved the spectrum broadening down to 780 nm, which overlaps with the wavelength of Rb atom transition. This is a great advantage of this study.

The article is composed by two parts (before and after line 208). The first half discusses on the dissipative Kerr soliton generation based on careful dispersion engineering of a SiN microring. A broad bandwidth comb is generated with 2-soliton state. Besides, octave spanning spectrum is obtained with 6-soliton states. The latter half is more interesting, which uses the avoided mode crossing to obtain local anomalous dispersion and obtain single dissipative Kerr soliton with 1060 nm (1064 nm?) pump.

I found this study to be important, because the soliton generation in visible wavelength may be one of the milestone for the micro-comb community. However, I have some reservation on accepting this paper because of the following three reasons. First, the octave spanning comb is achieved by the dispersion engineering of SiN waveguide, of which approach is not really new. Dispersion engineering of SiN waveguide has been well studied by Lipson & Gaeta's group. (May be not at this wavelength regime but, as far as I understood, the approach is the same). Second, the obtained state is not a single soliton state (in Fig. 3). The authors must pump the system at a high power level in order to obtain broad bandwidth output. However the strong pump results in high-order soliton (6-soliton) state. If this understanding is correct, it remains a big challenge to obtain octave spanning "single" soliton (1-FSR) state. Third, the authors claim that they generate single soliton pulse in Fig. 4. May be they are right, but the data is not sufficient to convince their claim. It would be nice if we can see that the pump is indeed at the red detuning side (soliton state), in order to proof that the state is indeed a soliton state.

Additional comments:

1. Related to the previous comment, the main weakness of this paper is that the effect of the careful dispersion engineering is not clearly presented. It would be nice if the authors can present a set of dispersions and the obtained spectra for different samples with different geometries. By presenting such data, the importance of the dispersion engineering should become clear. I understand the experiment is always difficult, but why not with simulations?

2. The waveguide the authors uses is multimode. This means that avoided mode crossings are present. Although the authors uses this phenomena in the latter half of this paper, it should be a problem in the first half. Often avoided mode crossing disturbs the dispersion profile and prevents the generation of broad bandwidth comb. So other researchers have tried to suppress the higher-order modes (i.e. in Ref. 9). I am wondering why the local disturbance of the dispersion does not affect the generation of the comb shown in Fig. 2 and 3, in their case.

3. Please make a comparison between Ref. 58. If I understand correct, the system used in this paper has lower Q (because it is SiN), but it is much smaller in size. Is it advantageous over the system used in ref. 58? Also, the FSR is much larger than that used in ref. 58. A large FSR should allow the authors to obtain broader bandwidth, which I thought to be an advantage. However, it is not possible to obtain a beat signal with such broad FSR, which is usually needed for the stabilization of the system (i.e. disadvantage). Please provide fair information on the pros & cons of the system (i.e. tradeoffs).

4. What is the reason behind that you only need standard wavelength sweeping approach in order to obtain soliton state (instead of using power kicking and frequency modulation method). Please discuss.

5. What is the pump wavelength? I guess it is around 1050 nm, but it is written as 1064 nm in line 70, and shown as 1060 nm in Fig. 1(a). Please clarify.

6. Some of the reference is not in a proper format, and I had difficulty to find the paper (i.e. Ref. 34 vol & page # are missing)

Reviewer #3 (Remarks to the Author):

The authors reported coherent soliton comb generation by pumping SiN microresonators in the 1 μm range. As has been pointed out in the paper, the generated frequency combs cover the biological imaging window and can be very attractive for biological applications. Furthermore, octave-spanning comb spectrally overlapping with Alkali atom transitions was also demonstrated. The results are impressive. My only major concern is related to the interpretation of mode-crossing induced combs as solitons (Figure 4). Note that pulse-like solutions are not necessarily solitons. Modulational instability combs can also be low-noise and pulse-like. Bright cavity solitons are localized pulses sitting atop a CW background. There can be multiple solitons coexisting in the cavity with random locations (or regulated as soliton crystals). No evidence of such localization is presented in the paper. No measurement of the time-domain waveform was performed. Actually even dark-pulse comb can have a rather smooth spectrum (see doi:10.1038/lsa.2016.253). The double resonance measurement is not sufficient to prove soliton formation. The authors should be more cautious to declare that these combs are solitons.

The following are some minor points that should be taken care of:

1. Page 5, line 131. The meaning of kappa is not explained.

2. Page 6, line 145. Fig. 2(c) should be Fig. 2(d).

3. Figure 2 title, line 8. (c) Heterodyne beatnote of the soliton comb line around 1050 nm with a second CW laser. "With" is missing.

4. In Fig. 2(a), the 3-soliton state is transferred to a 2-soliton state with pump back tuning. Can a single-soliton state be obtained if the pump is further back tuned? Can the authors comment on that?

5. In Fig. 1(d), the TM₀₀ modes seem very weak. But in Fig. 1(e), the extinction ratio of TM₀₀ looks much larger than that in Fig. 1(d). A brief comment would be helpful to avoid confusion.

I would suggest its publication in Nature Communications if the problems above have been properly addressed.

This paper written by Maxim et al reports on the dissipative soliton on silicon nitride microring for NIR and visible wavelength. This is a good integrated platform to generate comb with biological range. The octave spectrum is suitable for the stabilization of the comb by $f-2f$ and some short wavelength comb lines benefit the optical clock based on the Rb transition. But I think the properties of the comb in this paper do not reach the Nature communications level:

1. In Figure 1d, the transmission is not good, especially for TM mode. It means the coupling condition is not suitable for TM mode and pump power coupled into the cavity is low. So it is not best choice that the author select TM mode as pump.

- We thank the reviewer for his comment. The transmission shown in Fig. 1d is measured with a broadband calibrated laser frequency sweep, where the TM polarization was set at the start of the scan around 1030 nm (outside of the region shown in the figure) and was not adjusted during the sweep. Given that the laser light propagates through a significant amount of circularly folded non-PM fiber before entering the on-chip bus waveguide, the polarization plane can experience different amounts of rotation depending on the wavelength [Frins E. & Dultz W., JoLT 15(1), 1997]. Due to this reason, initial TM polarization coupled to the device is not maintained at different wavelengths, and appeared to be TE at some of them (as for example can be seen for the resonance around 1067.5 nm). This explains why adjacent modes of the 1-THz resonator are coupled differently over the range presented in the figure.

Figure R1. Resonance of the fundamental TM mode in 1-THz resonator at a wavelength of 1069.9 nm

On the other hand, for the soliton generation experiments presented in the other parts of the manuscript (Fig.2 - Fig.4), the polarization was properly adjusted each time for certain pumped resonances (whether it belongs to TE or TM fundamental mode families) beforehand.

In order to address the reviewer's concern about the low transmission of the TM mode, we first would like to draw his attention to Fig.1 (e) of the original manuscript, where the transmission traces for TE and TM fundamental modes are measured with properly adjusted polarizations of the input laser and demonstrate good bus-resonator coupling. Moreover we made an additional measurement of a similar microresonator on the same chip having different bus-resonator coupling gap distance, where the TM mode is coupled almost critically (see Fig. R1 of this reply).

We also want to explain our choice of the fundamental TM mode family instead of TE for dissipative Kerr soliton generation (DKS): Even though the TE mode family in our resonators is slightly better in terms of coupling to the bus waveguide and Q -factor, it has normal group velocity dispersion (GVD). This does not permit us to directly generate DKS, as they in general require anomalous GVD. On the other hand, in the latest section of our manuscript we show that TE mode can still be used for bright soliton generation, which was achieved via avoided-modal-crossing-induced local dispersion deviations, turning the GVD from normal to anomalous.

We have added the explanation of the different coupling of TE and TM fundamental mode families over the measurement range in Fig.1(d) to the “Design and Characterization” section of our manuscript. We have also mentioned there that the TE fundamental mode has normal GVD for the fabricated geometries.

2. In Figure 2, authors got the three and double pulses, not single pulse. This kind of comb has less comb lines and is not best source for bio-spectroscopy cause the FSR is already large.

- We have made additional experiments with the device presented in Fig.2 and generated a single soliton state via the backward tuning from a multiple –soliton state. We have updated Fig.2, and included the new data, which contain optical spectra of two- and single solitons states, fit of the single soliton state with a sech^2 spectral envelope, as well as the response measurements in all obtained states to prove the DKS formation.

3. In Figure 3a, although authors obtained octave spectrum, but both side comb lines are very weak and less. The $f-2f$ is very difficult and low efficient. According to the spectrum shape it is like breather soliton, not single pulse.

- We agree with the reviewer that the power in the wings of the soliton comb shown in Fig.3 is rather low, and *direct* $f-2f$ self-referencing using comb lines can be very challenging. Despite this fact, we strongly believe that this comb can still be self-referenced.

First, the actual power of the comb wings is higher by at least 4-6 dB from what is shown in the figure as a result of the fixed attenuation purposely introduced to prevent damaging of the optical spectrum analyzer.

Second, even if the power in the comb wings is low, one can still employ transfer lasers to implement $f-2f$ scheme for self-referencing. This approach was already successfully demonstrated by our group with $2f-3f$ scheme in similar Si_3N_4 -integrated microcombs operating at telecom wavelength [Brasch V. *et al*, Light: Science & Applications, 6, e16202 (2017)], and recently reproduced with $f-2f$ scheme in 1-THz resonators for on-chip optical frequency synthesizer [Spencer *et al*, Arxiv: 1708.05228]. In the first work, the power of the comb lines used for self-referencing was comparably low to our case (< -50 dBm), which however still allowed to measure f_{CEO} and achieve the stabilization. Third, in order to self-reference the soliton comb from the present work, one can also employ $2f-3f$ scheme using the lines with higher average power, reaching -20 dBm.

Regarding the second reviewer’s note – the state shown in Fig.3 is indeed not

a single soliton, but represents soliton crystal state with multiple temporally-arranged intracavity DKS [Cole *et al*, Nat. Phot. 11, 671–676 (2017)]. The state was quite detuned from the breathing region and existed in the region of stable stationary DKS. Moreover, low-frequency intensity noise measurements didn't show any trace of breathing [Lucas *et al*, Nat. Comm. 8 (736), 2017]. Therefore, in our opinion it is unlikely a breathing state.

4. The concept of hybridized mode is new, but it is difficult to control dispersion compared with mode interaction [Nat. Photon. 9, 594–600 (2015)].

- We have to admit that reviewer's concern regarding the dispersion is not entirely clear to us. The overall dispersion of the microring resonator in principle can be easily controlled with a high level of precision by adjusting the geometry of the waveguide and it is quite immune to the fabrication errors within a 5-nm-level. This was demonstrated for the DKS at the wavelengths of 1550 and 1300 nm [Pfeiffer *et al*, Optica 4, 7, (2017)]. On the other hand, the exact spectral positions of modal crossings are more difficult to predict, because they are more sensitive to the fabrication process variations.

Nevertheless the local dispersion tuning via the modal crossings, shown in the last part of our manuscript, can still be controllably used, because the position of crossings are thermally tunable as shown by Xue *et al* [Xue *et al*, L&PR 9(4), 2015].

So unfortunately this manuscript is not recommended at its present form and authors can design new chip again and optimize the properties of the comb.

- In the revised version of our manuscript we have significantly extended our results by (i) generating single soliton state, obtained through the backward tuning from multiple-soliton states; (ii) experimentally demonstrating the possibility of dispersion engineering at the wavelength of 1 μm , which results matched well to the FEM-based dispersion simulations for our waveguides; (iii) implementing additional measurements of the soliton states obtained in hybridized microresonator modes and proving that they indeed are bright dissipative Kerr solitons. We believe that with all new contributions, the current version of the manuscript properly addresses the reviewer's concerns and represents solid and complete research work.

This paper describes about the experimental demonstration of octave spanning soliton generation centered at 1050 nm wavelength. It is based on their previous study (Ref. 34), but extending their wavelength to 1050 nm regime by conducting careful dispersion engineering. Due to the tradeoff between the small FSR and large bandwidth, the use of a small cavity with large FSR of 1-THz allows the authors to obtain broad spectrum width, compare to the similar work conducted by Vahala et al. (ref. 58) that uses a cavity with 40-GHz FSR. As a result of the large bandwidth, this work achieved the spectrum broadening down to 780 nm, which overlaps with the wavelength of Rb atom transition. This is a great advantage of this study.

The article is composed by two parts (before and after line 208). The first half discusses on the dissipative Kerr soliton generation based on careful dispersion engineering of a SiN microring. A broad bandwidth comb is generated with 2-soliton state. Besides, octave spanning spectrum is obtained with 6-soliton states. The latter half is more interesting, which uses the avoided mode crossing to obtain local anomalous dispersion and obtain single dissipative Kerr soliton with 1060 nm (1064 nm?) pump.

I found this study to be important, because the soliton generation in visible wavelength may be one of the milestone for the micro-comb community. However, I have some reservation on accepting this paper because of the following three reasons.

1. First, the octave spanning comb is achieved by the dispersion engineering of SiN waveguide, of which approach is not really new. Dispersion engineering of SiN waveguide has been well studied by Lipson & Gaeta's group. (May be not at this wavelength regime but, as far as I understood, the approach is the same).

- The reviewer is correct that the dispersion engineering approach is indeed not new, and was demonstrated earlier at different wavelengths: 1550 nm, 1300 nm and even 1064 nm in optical microresonators [Okawachi *et al*, OL 39, 12 (2014); Saha *et al*, OE 20, 24, (2012); Pfeiffer *et al*, Optica 4, 7, (2017)]. However, the two main concerns that we focused on addressing in this work regarding the dispersion engineering were: (i) whether it is possible to engineer the dispersion in order **to achieve the DKS formation at around 1- μ m**, and (ii) whether it is possible to obtain an octave-spanning spectrum of the microcomb at this wavelength. The latter is typically achieved via the soliton dispersive wave emission [Brasch *et al*, Science, 351, 6271 (2016)], which can be excited by careful dispersion engineering. Such questions related to the DKS formation were never experimentally addressed in previous works.

In the present work, we demonstrated that despite increased normal material GVD of the silicon nitride platform for shorter wavelength, 1- μ m DKS can still be successfully generated, as well as the octave spanning soliton states via dispersive wave formation. To demonstrate that the dispersion was indeed consistently tuned via the waveguide geometry, we added to the supplementary information experimental optical spectra of devices with different waveguide widths (Section 1 of the Supplementary information). We also would like to draw the reviewer's attention to the additional

dispersion simulation data that we added to the main manuscript (Fig.1(c)) and to the Supplementary information, showing precise matching with the experiment.

2. Second, the obtained state is not a single soliton state (in Fig. 3). The authors must pump the system at a high power level in order to obtain broad bandwidth output. However the strong pump results in high-order soliton (6-soliton) state. If this understanding is correct, it remains a big challenge to obtain octave spanning "single" soliton (1-FSR) state.

- The state shown in Fig.3 is not a single soliton state, but represents a soliton crystal, which we estimate to consist of 6 almost equally spaced solitons. An important feature of this state is that its comb lines have a lot more power in comparison with the single soliton state that can be potentially obtained in this device at the same pump power of the driving laser. The reason is in the high level of temporal ordering of DKS pulses inside the cavity, which due to the interference results in the enhancement of each 6th comb line of the soliton state in the frequency domain. In combination with higher intracavity power (in our case it's also higher by 6 times, because cavity contains 6 DKS pulses) it gives a significant enhancement by more than 16 dB for each 6th line of the soliton state spectrum. This effect can be also noted by comparing power "gap" between the pump and the strongest soliton line for single DKS state in Fig.2 (c), which is more than 20 dB, and similar value for soliton crystal state in Fig.3 (a) - only 5 dB. The reviewer is correct that in order to obtain a single soliton state from this soliton crystal state we need to use higher pump powers, but the approach is trickier than just generating another state at high pump power. As experimentally demonstrated in our recent work, higher pump powers can enable soliton switching from soliton crystals down to a single soliton state [Karpov *et al*, CLEO 2017]. There, however, it is necessary to transfer the initial soliton crystal state from low pump powers to the relatively high ones, while maintaining the state in a stable regime. Due to the limitation on available power at 1- μ m in our experiment we indeed were unable to demonstrate such switching so far, but we see no obstacles to achieve it with higher pump powers \sim 5W. As an alternative solutions for the single-soliton generation, one can also use the recently suggested SSB-SC technique for fast pump frequency sweeping that allowed to generate single soliton states apart from thermal and other instabilities [Stone *et al*, CLEO (2017)]

3. Third, the authors claim that they generate single soliton pulse in Fig. 4. May be they are right, but the data is not sufficient to convince their claim. It would be nice if we can see that the pump is indeed at the red detuning side (soliton state), in order to proof that the state is indeed a soliton state.

- We thank the reviewer for the feedback. We indeed agree that our claim regarding the generation of dissipative Kerr solitons in hybridized modes should be better supported. In order to address this concern an additional measurements of this state generated in hybridized mode were carried out (see Supplementary information, section 2): we traced the evolution of the pump-modulation-based system response measurements Guo, Karpov, Lucas *et al*, Nat.Phys, 13 (2017), as well as changes in the optical spectrum and intensity noise of the transmitted light while applying pump backward tuning

(pump laser is tuned towards shorter wavelength). First, as mentioned in the main manuscript, we observed that the system response of the initial state has a characteristic two-resonance shape, which is attributed to the formation of soliton pulses inside the cavity [Guo, Karpov, Lucas *et al*, Nat.Phys, 13 (2017)] and consists of two contributions: from CW-background (so-called C-resonance) and from soliton pulses (S-resonance). Tracing the evolution of both contributions in the response while tuning backward, one can see (Fig.S2 of the supplementary information) that their behavior exactly repeats the behavior of dissipative Kerr solitons: the C-resonance is shifting towards smaller frequencies when the detuning is decreased, and at the merging with S-resonance the system experiences a sudden switch to a comb state with a structured optical spectrum and a significant amount of intensity noises (see Fig.S2 in the supplementary information). Such evolution and switching behavior are a strong evidence of the bright DKS state in the beginning of this measurement (and as a required condition for that – red-detuned pump position). Moreover, similar behavior was also demonstrated in the first part of our manuscript, where the soliton state with smaller anomalous GVD showed similar evolution of the system response, when decreasing the detuning (see Fig.2 (d) of the main manuscript).

We have included the new data in the supplementary information (SI) that we added to the revised manuscript submission (see Section 2 of the SI). We also would like to note, that we avoided any claim regarding the number of pulses in this DKS state, because we didn't have an unambiguous proof of a single soliton state.

Additional comments:

1. Related to the previous comment, the main weakness of this paper is that the effect of the careful dispersion engineering is not clearly presented. It would be nice if the authors can present a set of dispersions and the obtained spectra for different samples with different geometries. By presenting such data, the importance of the dispersion engineering should become clear. I understand the experiment is always difficult, but why not with simulations?

- We thank the reviewer for this feedback. We agree that experimental results showing the dispersion engineering are necessary to prove our claims.

Unfortunately, it is quite challenging to directly measure dispersion of 1-THz resonators due to large mode spacing and low dispersion values of our devices, because one would had to precisely measure resonance frequencies within a very broad range, covering several hundreds of nanometers just to extract D_2 dispersion parameter [Del'Haye, Nat. Phot, 2009]. Instead of doing this, one can measure the spectral locations of the phase-matching-induced enhancements of comb lines in the noisy comb states, which correspond to $D_{\text{int}}/2\pi = 0$ and provide a rough estimation for the position of soliton dispersive wave (DW) for a given geometry [Brasch *et al*, Science, 351, 6271 (2016)]. The positions of such phase-matched regions depend strongly on dispersion parameters and are easily tracked in experiment, allowing for convenient comparison to simulations. It is important, however, to highlight here that the generation of solitons is not needed for these measurements, and in fact may even give improper results, because the exact position of DW in a soliton state is slightly shifted from the phase-matched region where $D_{\text{int}}/2\pi = 0$ [Cherenkov *et al*, Phys. Rev. A 95, 033810 (2017)]. We also note, that a similar approach for the demonstration of

dispersion engineering in the silicon nitride platform was recently used for 1300 nm wavelengths [Pfeiffer *et al*, *Optica*, 4, 7, (2017)].

Following the reviewer's suggestion, we have experimentally measured optical spectra of comb states in different devices and added the results to the Supplementary Information (see Section 1 of SI). We used three samples with waveguide widths of 1.3, 1.35 and 1.4 μm and the height of 0.74 μm . The experimentally obtained position of the phase-matching region was shifting from 830 to 875 nm as the waveguide width was increased. These results are reproduced well in FEM-based dispersion simulations, when the experimental values for waveguide height and widths were used, and the sidewall angle was set to 77°.

In addition to the Section 1 of the SI, we also updated Figure 1(c) of the main manuscript to show how the dispersion of the resonator is affected by the waveguide height and width, and, in particular, how the position of the phase-matched regions ($D_{\text{int}} = 0$) are shifted when the geometry is varied. These results show the promising possibility to engineer the optical spectrum with two phase-matching regions, which will result in the DKS states with two dispersive waves.

As the last note here we want to mention that in experiments for octave-spanning DKS states (Figure 3), the device with the furthest position of the DW among available ones was employed.

2. The waveguide the authors uses is multimode. This means that avoided mode crossings are present. Although the authors uses this phenomena in the latter half of this paper, it should be a problem in the first half. Often avoided mode crossing disturbs the dispersion profile and prevents the generation of broad bandwidth comb. So other researchers have tried to suppress the higher-order modes (i.e. in Ref. 9). I am wondering why the local disturbance of the dispersion does not affect the generation of the comb shown in Fig. 2 and 3, in their case.

- We agree with the reviewer that the result seems counter-intuitive, as it is demonstrated that modal crossings, apart from their unwanted effects on DKS states revealed in previous works, can represent a precious way to tune dispersion, and, importantly, *enable* DKS formation.

First, we note, that the existence of avoided modal crossings (AMX) in our microresonators does affect the system and in particular the dynamics of solitons, which we indeed can observe. Nevertheless, the effect of AMXs is *not strong enough* to completely inhibit the DKS formation. In order to better explain this, it is convenient to use the phenomenological two-parametric model of the AMX, introduced by Herr *et al* [Herr *et al*, PRL 113 (123901), 2014]:

$$\omega_{\mu} = \omega_0 + \mu D_1 + \frac{1}{2} \mu^2 D_2 + \frac{a/2}{\mu - b - 0.5},$$

where a defines the maximum mode deviation in the crossing point, and b - how far

Figure R2. Schematics of the two-parametric model of AMX (adapted from [Herr *et al*, PRL, 113 (2014)])

the crossing is located from the pump line (see Fig.R2 of this reply). As it was shown by Herr *et al.*, when AMX is strong (a is large), or stays very close to the pump (b is small) the soliton generation can indeed be inhibited due to catastrophic pulse interaction with a strongly modulated background resulting from the AMX. In other cases, when the modal crossing is far from the pump (b is large), or the AMX is weak (a is small), the effect of the modal crossing on the DKS dynamics is moderate: solitons *can still be generated*, and they are able to coexist with modulated CW-background. In the latter case, AMXs result in the reduced soliton existence range (in comparison with no-AMX case) and the appearance of the soliton inter-mode breathing [Guo *et al.*, arXiv:1705.05003] at different values of pump power and detuning, where the system is expected to be in a stable *stationary* DKS regime.

In our work, we do observe the onset of the inter-mode breathing in DKS states (similar to [Guo *et al.*, arXiv: 1705.05003]) at certain values of pump power and pump-resonance detuning, which is the result of AMXs. We can also see that this effect may complicate the soliton switching process via the backward tuning: at certain pump powers the DKS state can experience strong breathing and be destroyed before switching in a single soliton state. Another effect of the AMX is the formation of a soliton crystal state shown in Figure 3 of the main manuscript. It results from AMX-induced modulation on the CW-background, which provides temporal ordering of the soliton pulses [Cole *et al.*, Nat. Phot. 11, 671–676 (2017)]. Nevertheless, in the samples used in both Fig.2 and Fig.3, the strength of AMXs was not enough to completely inhibit the soliton formation, or made them unstable – all measured states were operating stably for several hours.

In contrast, in the last part of our manuscript we directly use dispersion alterations of hybridized microresonator mode forming AMX, and show that such hybridized mode family can have strong anomalous GVD enabling the formation of dissipative Kerr solitons (see also updated figure 4 of the main manuscript for the scheme of the described phenomenon).

3. Please make a comparison between Ref. 58. If I understand correct, the system used in this paper has lower Q (because it is SiN), but it is much smaller in size. Is it advantageous over the system used in ref. 58? Also, the FSR is much larger than that used in ref. 58. A large FSR should allow the authors to obtain broader bandwidth, which I thought to be an advantage. However, it is not possible to obtain a beat signal with such broad FSR, which is usually needed for the stabilization of the system (i.e. disadvantage). Please provide fair information on the pros & cons of the system (i.e. tradeoffs).

- There are indeed several pros and cons for both systems:
First, our silicon nitride devices have lower Q -factor of less than 1 million at 1060 nm (however, recent advances have revealed that intrinsic Q -factor of silicon nitride can be much higher, ca. 20 million [Li *et al.*, OE, 21, (2013)]), while in silica wedge-resonators the Q -factor can reach almost 100 million at the same wavelength. This greatly impacts the threshold for parametric oscillations and the power needed for DKS generation: even though the nonlinearity of silicon nitride is more than 10 times higher, and the mode volume of our resonators is smaller, the power threshold for DKS generation in silica wedge-resonators is still smaller by about two orders of magnitude.
Second, silicon nitride microring resonators are more flexible in terms of the available

free spectral ranges (FSR), which can span from 10^5 to 1000^5 of GHz, while for silica wedge-resonators the best Q -factors are available for $FSR < 50$ GHz. This can be considered as a disadvantage, especially for the spectroscopy applications where higher FSR are able to provide higher sampling speeds [Ideguchi *et al*, Nat. Phot. 502, 2013]. Higher FSR also allows an octave-spanning bandwidth to be achieved, which is particularly important for self-referencing.

Third, from the application perspective silicon nitride platform is considered to be more promising, because the devices are integrated (i.e. have integrated bus waveguides) and are oxide-cladded, which makes them immune to external impacts. Equally important, the photonic chip can be directly coupled to optical fibers and has well-defined coupling facets. In contrast, current realizations of wedge-resonators supposes the usage of external fiber coupling, which severely limit the system integration in other devices. Although Si_3N_4 waveguide coupling has recently been achieved, the key issues at hand remain efficient coupling to fiber from uncladded devices as well as integration with further functionality (e.g. heaters, PZT) and requirement of very thick oxides.

4. What is the reason behind that you only need standard wavelength sweeping approach in order to obtain soliton state (instead of using power kicking and frequency modulation method). Please discuss.

- In our opinion this fact is explained by the relatively low power of the pump laser which we use to excite soliton states. Low pump powers allow us to avoid the region of transient chaos in the system stability map [Leo *et al*, OL 7(21), 2013], when generating a soliton state with a standard tuning procedure (tuning the pump laser with *fixed* pump power from short to long wavelengths). This enables almost all transient intracavity pulses (existing at the last instance of the modulation instability) to be transferred to solitons, when the system enters the region of stable DKS. In this scenario intracavity power does not change significantly, and the thermal effects do not affect the pump-cavity detuning, maintaining the state in the stable regime. This fact allows us to tune in soliton states using slow tuning speeds or even manual tuning. In contrast, in our previous work on DKS in Si_3N_4 and MgF_2 at 1550 nm [Guo, Karpov, Lucas *et al*, Nat.Phys 13, (2017)], we used higher pump powers, and the systems experienced a large drop of intracavity power, when tuned in the stable DKS regime. In that case one had to use fast tuning speeds for soliton excitation to prevent significant change of the system's temperature after entering the stable DKS regime [Herr *et al*, Nat. Phot. 8, 2014].

We would like to provide the detailed investigation of this phenomenon and it's relation to the soliton formation in our forthcoming work on soliton crystals, where this effect plays a major role [Karpov *et.al*, CLEO 2017].

We have included a short explanation of the effect in the manuscript and added a reference to our work.

5. What is the pump wavelength? I guess it is around 1050 nm, but it is written as 1064 nm in line 70, and shown as 1060 nm in Fig. 1(a). Please clarify.

- We thank the reviewer for this note. The pumping wavelengths in our experiments

were within 1055-1070 nm, which was defined by the available range of our ytterbium-doped fiber amplifier. For the generation of single and multiple soliton states in the first part of the work we used pump wavelengths of around 1064.6 nm. For the generation of the octave-spanning DKS – the pump was at 1051.5 nm.

We have indicated the pump wavelength in the corresponding figure captions, and changed the wavelength at the line 70 to “1060 nm”.

6. Some of the reference is not in a proper format, and I had difficulty to find the paper (i.e. Ref. 34 vol & page # are missing)

- We thank the reviewer for pointing this out. The volume and page were added to the reference.

The authors reported coherent soliton comb generation by pumping SiN microresonators in the 1 μm range. As has been pointed out in the paper, the generated frequency combs cover the biological imaging window and can be very attractive for biological applications. Furthermore, octave-spanning comb spectrally overlapping with Alkali atom transitions was also demonstrated. The results are impressive. My only major concern is related to the interpretation of mode-crossing induced combs as solitons (Figure 4). Note that pulse-like solutions are not necessarily solitons. Modulational instability combs can also be low-noise and pulse-like. Bright cavity solitons are localized pulses sitting atop a CW background. There can be multiple solitons coexisting in the cavity with random locations (or regulated as soliton crystals). No evidence of such localization is presented in the paper. No measurement of the time-domain waveform was performed. Actually even dark-pulse comb can have a rather smooth spectrum (see doi:10.1038/lsa.2016.253). The double resonance measurement is not sufficient to prove soliton formation. The authors should be more cautious to declare that these combs are solitons.

- We thank the reviewer for this comment. We are sure that the observed states are bright dissipative Kerr solitons (DKS) due to these several reasons:
 - First, we experimentally verified (see Fig.4 of the manuscript) that the local group velocity dispersion (GVD) for the resonances of the TE mode family composing the modal crossing is strongly anomalous, which is a precondition for the generation of *bright* soliton pulses. It should also be noted that we don't observe the "locking" of the primary sidebands to a certain spectral positions (defined by modal crossings), which was demonstrated to be a sign of AMX-induced modulation instability in the case of normal GVD [Xue *et al*, Nat. Photonics 9, (2015)].
 - Second, in our previous work [Guo, Karpov, Lucas *et al*, Nature Physics, 13 (2017)] we have theoretically and experimentally demonstrated that the dual-peak response obtained in the response measurements is an unambiguous evidence of DKS formation inside the cavity. Moreover, we complemented our present results with additional measurements (see Section 2 of Supplementary information (SI) for details), where the evolution of the system response was traced in the hybridized-mode soliton state, when the pump backward tuning is applied. The measured evolution of the dual-peak response, and its change to a single-resonance peak when switching to a high-noise comb state are qualitatively very similar to the behavior of DKS pulses and switching to the noisy Kerr comb, demonstrated earlier in different microresonator platforms [Guo, Karpov, Lucas *et al*, Nature Physics, 13 (2017)].
 - Third, while generating the states in hybridized modes by pump tuning, we experimentally observe a very typical excitation path for DKS states, which starts with modulation instability (appearance of the primary comb lines), followed by chaotic modulation instability with high broadband intensity noise and very structured spectrum, and finishes in a low-noise state with a smooth 1-FSR spectrum which also exhibits the characteristic DKS dual-peak system response.
 - Fourth, in order to unambiguously demonstrate the soliton behavior, we have made an additional measurements by generating this state in one of the hybridized

modes and tracing the evolution of the pump-modulation-based response measurements, while applying pump backward tuning (pump laser is tuned towards shorter wavelength). At the same time we recorded changes in the optical spectrum and intensity noise of the transmitted light during the same process. It was observed, that the characteristic *C*- and *S*-resonances, corresponding to the cavity and soliton contributions to the system response behave exactly as the ones of bright dissipative Kerr solitons [Guo, Karpov, Lucas et al, Nat. Phys, 2017]: the *C*-resonance is shifting towards smaller frequencies when the detuning is decreased, and at the merging with the *S*-resonance the system experiences the sudden switch to a comb state with a structured optical spectrum and a significant amount of intensity noises (see Fig.S2 in the supplementary information). Such switching behavior provides strong evidence of DKS formation in the beginning of this measurement.

In conclusion we are sure the state is a DKS state, as it has multiple similarities to DKS behavior, including excitation dynamics, low-noise operation and system response dynamics.

We have added all the relevant data to the SI of our manuscript in order provide sufficient proofs for our claim on the observation of DKS in hybridized modes.

The following are some minor points that should be taken care of:

1. Page 5, line 131. The meaning of kappa is not explained.
2. Page 6, line 145. Fig. 2(c) should be Fig. 2(d).
3. Figure 2 title, line8. (c) Heterodyne beatnote of the soliton comb line around 1050 nm with a second CW laser. "With" is missing.

- We thank the reviewer for these remarks. All corresponding changes have been implemented in the manuscript.

4. In Fig. 2(a), the 3-soliton state is transferred to a 2-soliton state with pump back tuning. Can a single-soliton state be obtained if the pump is further back tuned? Can the authors comment on that?

- The pump backward tuning can indeed be used in our system to switch to the single soliton state from higher-number soliton states. We demonstrated this in additional experiments and included new data in figure 2 of the main manuscript. Fig.2 now contains the spectra of two- and single soliton states obtained in the same device, as well as response measurements for them.

The backward tuning for the current devices does not work as perfect as in our previous work on soliton switching [Guo, Karpov, Lucas *et al*, Nat. Phys 13 (2017)]. We assume that the number of avoided modal crossings (AMX) in our multimode microresonator system can induce inter-mode breathing instabilities and shorten the soliton existence range of soliton states. In this case, the backward tuning might be complicated at certain pump powers where the effect of AMX is most prominent [Guo *et al*, arXiv: 1705.05003]. Firstly, because the multiple-soliton states can decay completely before reaching the switching point due to strong breathing or, secondly, they can lose several solitons at once which causes a strong thermal shift of the cavity

resonance, changing the detuning, and shifting the system out of the stable soliton region causing decays [Guo, Karpov, Lucas *et al*, Nat. Phys 13 (2017)].

5. In Fig. 1(d), the TM₀₀ modes seem very weak. But in Fig. 1(e), the extinction ratio of TM₀₀ looks much larger than that in Fig. 1(d). A brief comment would be helpful to avoid confusion.

- We thank the reviewer for this question. We agree that we had to be clearer on this point. The explanation is that the measurement was implemented with a wide laser scan, where the TM polarization was set in the beginning of the scan (outside of the region shown in the figure) at 1030 nm, and was not adjusted afterwards. Since the laser light propagates through a significant amount of circularly folded non-PM fiber before coupling to the device, the polarization plane can experience different amounts of rotation depending on the wavelength [Frins E. & Dultz W., JoLT 15(1), 1997]. This can explain the fact that adjacent modes of the 1-THz resonator are coupled differently over the range presented in the figure – the initial TM polarization coupled to the device is not maintained at the other wavelengths, and appeared to be close to TE at some of them (as for example can be seen for 1067.5nm). We have added the explanation of this confusion to the manuscript.

I would suggest its publication in Nature Communications if the problems above have been properly addressed.

REVIEWERS' COMMENTS:

Reviewer #1 (Remarks to the Author):

Thanks for the professional and clear reply. The 0.7-1.4 microns comb is very important for the bio-window. But recently Vahala's group just reported the similar work about 0.7 and 1.064 microns Kerr comb using the microdisk cavity (Towards visible soliton microcomb generation, Nat Commun. 2017; 8: 1295.). The Q factor reach 80 million and FSR is about 20GHz. It means that this kind of Kerr comb need lower pump power and provide good resolution for the spectroscopy.

This manuscript used the silicon nitride microring to realize the 1064 Kerr comb and it is more suitable for the integrated microcomb. But I think this is only technology improvement. So I suggest that author can submit this manuscript to optics journal, like Optica.

Reviewer #2 (Remarks to the Author):

Previously, I raised three concerns:

1. The approach is not new.
2. The obtained state is not a single soliton.
3. Kerr solitons in hybridized modes should be better supported.

In the revised manuscript, the authors added a section in the supplemental to provide better support for the proof #3. Now the concern is resolved. However, I am yet not fully satisfied with the answers given for #1 and #2. Regarding #1, I agree to the authors that the result is of importance (having achieved the DKS formation at around 1- μm), but this does not mean that the approach is new. Although the authors discuss on some methods that may enable single soliton state (#2), the experiment is not performed.

On the other hand, I agree to the authors that the obtained result is of importance; namely the soliton generation in visible wavelength. Considering this fact and those #1 and #2, I would say that the paper is at the borderline; or slightly above.

I am satisfied with other revisions (additional questions) made by the authors.

Reviewer #3 (Remarks to the Author):

The authors have properly addressed all my questions in their revised manuscript. I think it is suitable for publication now.

Reviewer #1

Thanks for the professional and clear reply. The 0.7-1.4 microns comb is very important for the bio-window. But recently Vahala's group just reported the similar work about 0.7 and 1.064 microns Kerr comb using the microdisk cavity (Towards visible soliton microcomb generation, Nat Commun. 2017; 8: 1295.). The Q factor reach 80 million and FSR is about 20GHz. It means that this kind of Kerr comb need lower pump power and provide good resolution for the spectroscopy.

This manuscript used the silicon nitride microring to realize the 1064 Kerr comb and it is more suitable for the integrated microcomb. But I think this is only technology improvement. So I suggest that author can submit this manuscript to optics journal, like Optica.

- We thank the reviewer for his note. First, we would like to point out, that both works (the current one and the work from Vahala's group) have been conducted *concurrently*. It should also be noted that we were the first to report our results, which has been done at the Postdeadline session of CLEO 2017 [Karpov et al. "Chip-scale dissipative-Kerr-soliton-based frequency combs driven with 1 μ m source", Conference on Lasers and Electro-Optics (2017)], and therefore we were the first to publish them in the corresponding conference proceedings. To avoid any confusions on this point regarding priority we have changed the author's note at the end of our manuscript. Second, we also would like to draw reviewer's attention to the difference between two platforms presented in both works. Even having higher quality factors and smaller FSR, the current microdisk platform have several significant limitation. First, it can hardly be truly chip-integrated because of the tapered-fiber-coupling, which significantly reduces robustness of the system and narrows the range of potential applications. Second, a strong group velocity dispersion of the microdisk resonators does not allow for broadband comb formation, which is preferred for major biomedical applications, including CARS and OCT, and which is absolutely required for the self-referencing. Given that we have to admit that we do not share the reviewer's point of view that our work is only "a technological improvement", because in our opinion Si₃N₄ platform represents more promising candidate for the applications, and moreover required significant advances and efforts in the design, fabrication and dispersion engineering to achieve fully chip-integrated devices operating at 1 μ m wavelength.

Reviewer #2

Previously, I raised three concerns:

1. The approach is not new.
2. The obtained state is not a single soliton.
3. Kerr solitons in hybridized modes should be better supported.

In the revised manuscript, the authors added a section in the supplemental to provide better support for the proof #3. Now the concern is resolved. However, I am yet not fully satisfied

with the answers given for #1 and #2. Regarding #1, I agree to the authors that the result is of importance (having achieved the DKS formation at around 1- μm), but this does not mean that the approach is new. Although the authors discuss on some methods that may enable single soliton state (#2), the experiment is not performed.

On the other hand, I agree to the authors that the obtained result is of importance; namely the soliton generation in visible wavelength. Considering this fact and those #1 and #2, I would say that the paper is at the borderline; or slightly above.

I am satisfied with other revisions (additional questions) made by the authors.

- We thank the referee for his comments. As we already mentioned in our previous reply, we totally agree that the concept of tailoring the dispersion through the engineering of the waveguide geometry is not new. However we still would like to point out that such dispersion engineering is not a central idea of our manuscript, but just a mean of achieving the goal – broadband dissipative Kerr soliton states at 1 μm wavelength covering the biological imaging window.

Reviewer #3

The authors have properly addressed all my questions in their revised manuscript. I think it is suitable for publication now.

- We thank the reviewer for his feedback and contributions for improving our manuscript.